# Unified classification and risk-stratification in Acute Myeloid Leukemia

Yanis Tazi [1,2,3,4], Juan E. Arango-Ossa[1,2], Yangyu Zhou[1,2], Elsa Bernard [1,2], Ian Thomas[5], Amanda Gilkes[6], Sylvie Freeman [7], Yoann Pradat [1], Sean J. Johnson[5], Robert Hills[8], Richard Dillon [9], Max F. Levine [1], Daniel Leongamornlert [10], Adam Butler[10], Arnold Ganser[11], Lars Bullinger[12], Konstanze Döhner [13], Oliver Ottmann[6], Richard Adams [5], Hartmut Döhner [13], Peter J. Campbell [10], Alan K. Burnett [14], Michael Dennis[15], Nigel H. Russell[16,18], Sean M. Devlin [1,18], Brian J. P. Huntly [17,18] & Elli Papaemmanuil [1,2,18] ✉

Clinical recommendations for Acute Myeloid Leukemia (AML) classification and risk-stratification remain heavily reliant on cytogenetic findings at diagnosis, which are present in <50% of patients. Using comprehensive molecular profiling data from 3,653 patients we characterize and validate 16 molecular classes describing 100% of AML patients. Each class represents diverse biological AML subgroups, and is associated with distinct clinical presentation, likelihood of response to induction chemotherapy, risk of relapse and death over time. Secondary AML-2, emerges as the second largest class (24%), associates with high-risk disease, poor prognosis irrespective of flow Minimal Residual Disease (MRD) negativity, and derives significant benefit from transplantation. Guided by class membership we derive a 3-tier risk-stratification score that re-stratifies 26% of patients as compared to standard of care. This results in a unified framework for disease classification and risk-stratification in AML that relies on information from cytogenetics and 32 genes. Last, we develop an open-access patient-tailored clinical decision support tool.

[1]Computational Oncology Service, Department of Epidemiology & Biostatistics, Memorial Sloan Kettering Cancer Center, New York, NY, USA. [2]Center for Hematologic Malignancies, Memorial Sloan Kettering Cancer Center, New York, NY, USA. [3]Tri-Institutional Computational Biology and Medicine PhD Program, Weill Cornell Medicine of Cornell University and Rockefeller University, New York, NY, USA. [4]The Rockefeller University, New York, NY, USA. [5]Centre for Trials Research, School of Medicine, Cardiff University, Cardiff, UK. [6]Department of Haematology, School of Medicine, Cardiff University, Cardiff, UK. [7]Institute of Immunology and Immunotherapy, University of Birmingham, Birmingham, UK. [8]Nuffield Department of Population Health, University of Oxford, Oxford, UK. [9]Department of Medical and Molecular Genetics, King's College, London, UK. [10]Cancer, Ageing and Somatic Mutation Programme, Wellcome Sanger Institute, Hinxton, UK. [11]Department of Hematology, Hemostasis, Oncology, and Stem Cell Transplantation, Hannover Medical School, Hannover, Germany. [12]Department of Hematology, Oncology, and Tumorimmunology, Campus Virchow Klinikum, Berlin, Charité—Universitätsmedizin Berlin, corporate member of Freie Universität Berlin and Humboldt-Universität zu Berlin, Berlin, Germany. [13]Department of Internal Medicine III, Ulm University, Ulm, Germany. [14]Visiting Professor University of Glasgow, formerly Cardiff University, Cardiff, UK. [15]The Christie NHS Foundation Trust, Manchester, UK. [16]Department of Haematology, Nottingham University Hospital, Nottingham, UK. [17]Department of Haematology and Wellcome Trust-MRC Cambridge Stem Cell Institute, University of Cambridge, Cambridge, UK. [18]These authors contributed equally: Nigel H. Russell, Sean M. Devlin, Brian J. P. Huntly, Elli Papaemmanuil. ✉e-mail: papaemme@mskcc.org

Acute Myeloid Leukemias (AML) are overlapping hematological neoplasms associated with rapid onset, progressive and frequently chemoresistant disease[1,2]. Intensive chemotherapy and combination regimens have recently shown improvement in patient response[3,4], however, the risk of relapse-related mortality remains high[5]. At diagnosis, classification and risk-stratification are critical for treatment decisions[2–4]. Decisions on type of consolidation chemotherapy, timing of hematopoietic stem cell transplantation (HSCT) or eligibility for clinical trials[3], are evaluated on each patients' a priori likelihood of attaining complete remission (CR), the prospective persistence of measurable residual disease[6] (MRD), and the predicted likelihood of relapse or death[2].

As prospective sequencing is becoming routine during AML diagnosis, there is a need to understand the clinical relevance of molecular biomarkers in the context of established endpoints (i.e. MRD, CR, relapse). Translation of such findings into clinical practice warrants the development of evidence-based and dynamic clinical decision support tools that consider molecular and clinical biomarkers to inform optimal diagnosis and treatment decisions and improve patient outcomes[7].

To this end, gene mutations are being gradually incorporated into classification and risk-stratification guidelines for AML patient management[1,2]. However, with the exception of NPM1, CEBPA and provisionally RUNX1, the WHO[2016] classification is primarily reliant on cytogenetic findings[1,2]. Here, we incorporate data from 2113 representative AML patients enrolled in three UK-NCRI trials[8,9]. We study the relationships between genetic alterations, clinical presentation, treatment response and outcome to develop a framework that unifies diagnostic classification to risk stratification that results in significant improvement in predictive accuracy. Results were validated in an independent cohort of 1540 AML patients[10].

## Results
### Study participants
Study participants included 2113 AML adult patients enrolled in UK-NCRI trials[3,8,9] (training), which uniquely recruit up to 80% of UK patients fit for either intensive or non-intensive treatment and are therefore representative of the "real-world" patient population rather than studies limited by strict trial entry criteria. The majority (83%, n = 1755) were intensively treated[8,11,12] (median age = 56). Data from 1540 AML patients from the AML-SG[10] (median age = 50) with comparable molecular annotation at diagnosis were used as a validation cohort (Supplementary Table 1, Supplementary Data 1, Supplementary Fig. 1). Informed consent was obtained for all patients. Molecular assessment of UK-NCRI cohort included karyotypes[8,9], copy number alterations (CNA) and putative oncogenic mutations across the entire gene body of 128 genes implicated in myeloid neoplasia pathogenesis at diagnosis (Supplementary Data 2–4).

### Genomic landscape of AML
Mapping of recurrent cytogenetic abnormalities and gene mutations characterized 8,460 driver events in 98% of the UK-NCRI cohort (Supplementary Data 3, 4) Genotype and clinical relationships for 70 recurrent cytogenetic abnormalities and 84 genes were consistent with prior studies (Supplementary Figs. 3, 4; https://www.aml-risk-model.com/gene-panel). Detailed genotype and clinical relationships to include patterns of co-mutation, clinical and outcome correlates were evaluated for each of 70 recurrent (>1%) cytogenetic abnormalities and 84 genes with established role in AML pathogenesis (Supplementary Fig. 4; https://www.aml-risk-model.com/supplementary).

### Molecular classification in AML
Utilizing the WHO[2016] guidelines for AML classification, 49.6% (n = 1049) of UK-NCRI patients mapped to established WHO[2016] classes. Each class ranged in size from 0.4% to 31.5% (Supplementary Fig. 5).

Clustering analysis on the basis of cytogenetic and gene mutation findings identified 14 non-overlapping clusters classifying 92% (n = 1943) of patients (Supplementary Figs. 6, 7). These validate established WHO[2016] entities, resolve provisional subgroups[2,13,14] and determine previously uncharacterized molecular subgroups that describe 33.3% of AML patients (Supplementary Fig. 8). Each class is associated with distinct demographic and clinical parameters and in unison, explain the heterogeneity observed at diagnosis across age, peripheral blood and blast counts amongst AML patients (Supplementary Table 2).

Classes defined by cytogenetic alterations included entities defined by translocations and patients with complex karyotype (CK, ≥ 3 unbalanced abnormalities) (n = 217, 10.3%) with frequent involvement of TP53 mutations (n = 141, 65%)[14]. Consistent with prior studies, patients with CK were generally older (median diagnostic age = 62) and associated with adverse outcomes[10]. With the exception of mutations in TP53, which was mutated in 65% of CK cases, there was a paucity of other acquired mutations in this group. Unlike in MDS[15], the allelic state of TP53 (mono allelic or multi-hit) provided no further prognostic information in AML (Supplementary Fig. 9). A novel cytogenetic subgroup was defined by the presence of ≥1 trisomies (n = 237, 11.2%), frequently involving +8, +11, +13, +21 and +22 but no deletions. This group had infrequent involvement of TP53 (4%), and was associated with more favorable disease, even when ≥3 trisomies were present (Fig. 1a, b, Supplementary Fig. 10).

Patients with ≤2 aneuploidies (n = 233, 11%), enriched for "MDS-related"[16,17] cytogenetic abnormalities clustered with secondary AML type mutations (sAML)[16] such as SRSF2, SF3B1, U2AF1, ZRSR2, ASXL1, EZH2, BCOR, or STAG2, as well as novelly described here, RUNX1, SETBP1, and MLL[PTD] mutations. This represented the second largest cluster (28.4%, n = 601). Patients in this group were older (median diagnostic age 65.5 vs 56, p < 0.0001), with lower blast counts (median 51 vs 65, p < 0.0001) and higher incidence of antecedent hematologic disease (AHD) (32% vs 11.4% p < 0.0001) (Fig. 1c). Given the enrichment of MDS-related abnormalities and sAML like gene mutations, we name this cluster "sAML" per Lindsley et al [18]. Of prognostic importance, the association with adverse outcomes was specific to patients with ≥2 mutations (5-year survival rate = 16%), as compared to patients with a single gene mutation in a class-defining gene (5-year survival rate = 37%) (Fig. 1d, Supplementary Figs. 11, 12, Supplementary Table 2). Thus, we subdivided this cluster into secondary AML Like-1 (sAML1) defined by patients with single mutations (n = 100) and secondary AML Like-2 (sAML2) for patients with ≥2 class defining genes (n = 501) (Fig. 1d, Supplementary Figs. 11, 12). AHD was enriched in sAML2 and associated with even worse outcomes (p < 0.0001) (Supplementary Fig. 13). The provisional WHO entity defined by RUNX1 mutations (13.5%) spread across both sAML1 and sAML2 subgroups at similar frequencies and did not confer independent prognosis (Supplementary Fig. 14).

In the absence of classifying events (e.g CEBPAbi, t(8;21)), WT1 mutations defined a distinct cluster (n = 40, 2%) (Supplementary Fig. 15) characterized by few mutations, younger diagnostic age (median = 41), high white blood cell (WBC) counts and intermediate risk, unless co-mutated with FLT3[ITD] (Supplementary Fig. 15). We further validated the DNMT3A/IDH class in 1% of patients and demonstrate that this is a heterogeneous group (Supplementary Fig. 16)[14]. 6% of patients (n = 124) not clustering with any class were labeled as "molecularly Not Otherwise Specified" (mNOS) and 2% had no identifiable mutation (n = 46) in our panel (no mutations).

These findings informed a hierarchical classification that explicitly assigns 100% of patients into a molecular class (including mNOS and no-mutations) (Fig. 1e, Supplementary Fig. 17, https://www.aml-risk-model.com/supplementary). Patients in the sAML1, sAML2, WT1 and trisomy classes demarcated independent prognostic groups relative to ELN[2017] (Fig. 2a, Supplementary Fig. 18).

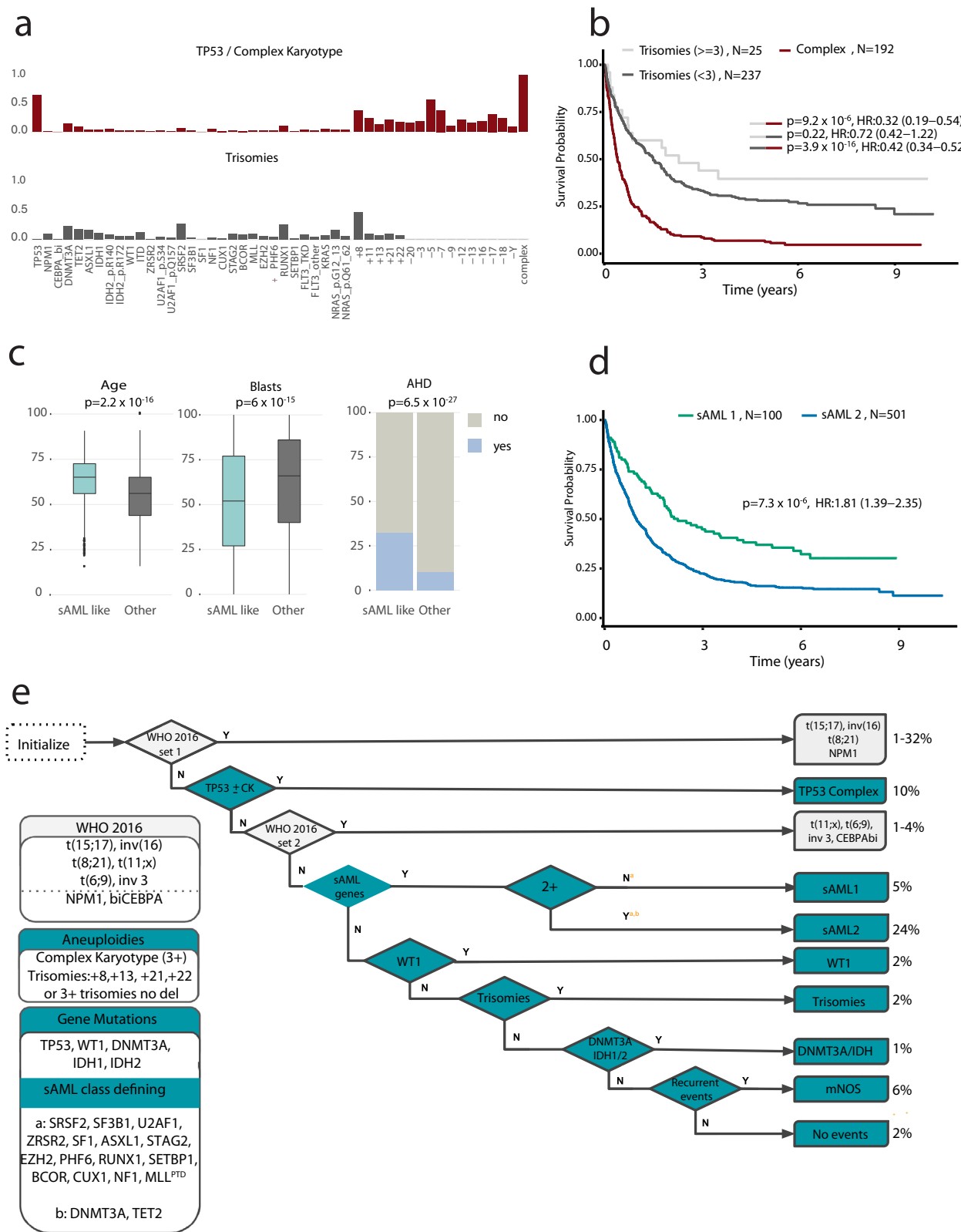

This is important as in the absence of risk stratifying biomarkers a significant proportion of patients in these newly defined groups were considered as intermediate risk AML (eg. 18% of patients in the AML2 class) (Fig. 2a). Patients with no-mutations had favorable outcomes and were distinct from intermediate-risk mNOS. This demonstrates that given a comprehensive workup, negative findings also provide relevant prognostic information (Figs. 1e and 2a, Supplementary Fig.

18). Proposed class associations were validated in AMLSG (Supplementary Fig. 19). As expected, non-intensively treated patients in the NCRI cohort were enriched in the TP53-CK and sAML2 groups. Nonetheless, the associations with adverse outcomes remained, as in the intensively treated subsets (Supplementary Fig. 20).

Notably, similarly to most signaling gene mutations (e.g. *NRAS*), *FLT3* mutations are present across classes. Thus, these mutations are

**Fig. 1 | Molecular classification in AML. a** Repartition of two patterns of chromosomal aneuploidies to include TP53 and complex and trisomies. The *y*-axis represents the fraction of patients carrying each driver event (on the *x*-axis) for each of the two subgroups (training, *n* = 2113). **b** Kaplan–Meier overall survival curves for overall survival curves for patients with trisomies (<3)(gray), trisomies(≥3)(lightgrey) and complex karyotype (burgundy) in the training cohort (*n* = 2113). Log-rank tests compared the survival distributions between complex and MDS related cytogenetics and between complex and trisomies not complex subgroups. **c** Comparison of age (years), bone marrow blasts (%) and AHD (antecedent hematologic disorder) distributions for sAML like subgroups (*N* = 601) to other AML in AML NCRI cohort (*N* = 1512). Two-sided *p*-values on the

boxplots used either a Wilcoxon rank-sum test or a Fisher's exact test. **d** Kaplan–Meier overall survival curves for the secondary AML like classes (sAML1 and sAML2) in the training cohort (*n* = 2113). Annotated *P*-values are from two-sided log-rank tests. **e** Hierarchical classification schema. Hierarchy rules for AML class assignment, biomarkers for hierarchy implementation and class range proportions. sAML2 comes before biCEPBA in the hierarchy (Supplementary Appendix for more details). WHO 2016 set1 and WHO 2016 set2 display classifications for more than one group. For those 2 specific boxes, we displayed range values representing the proportions of the smallest class and largest class in that subset. For all other sets, the values represent the proportion of patients in the cohort for that particular class.

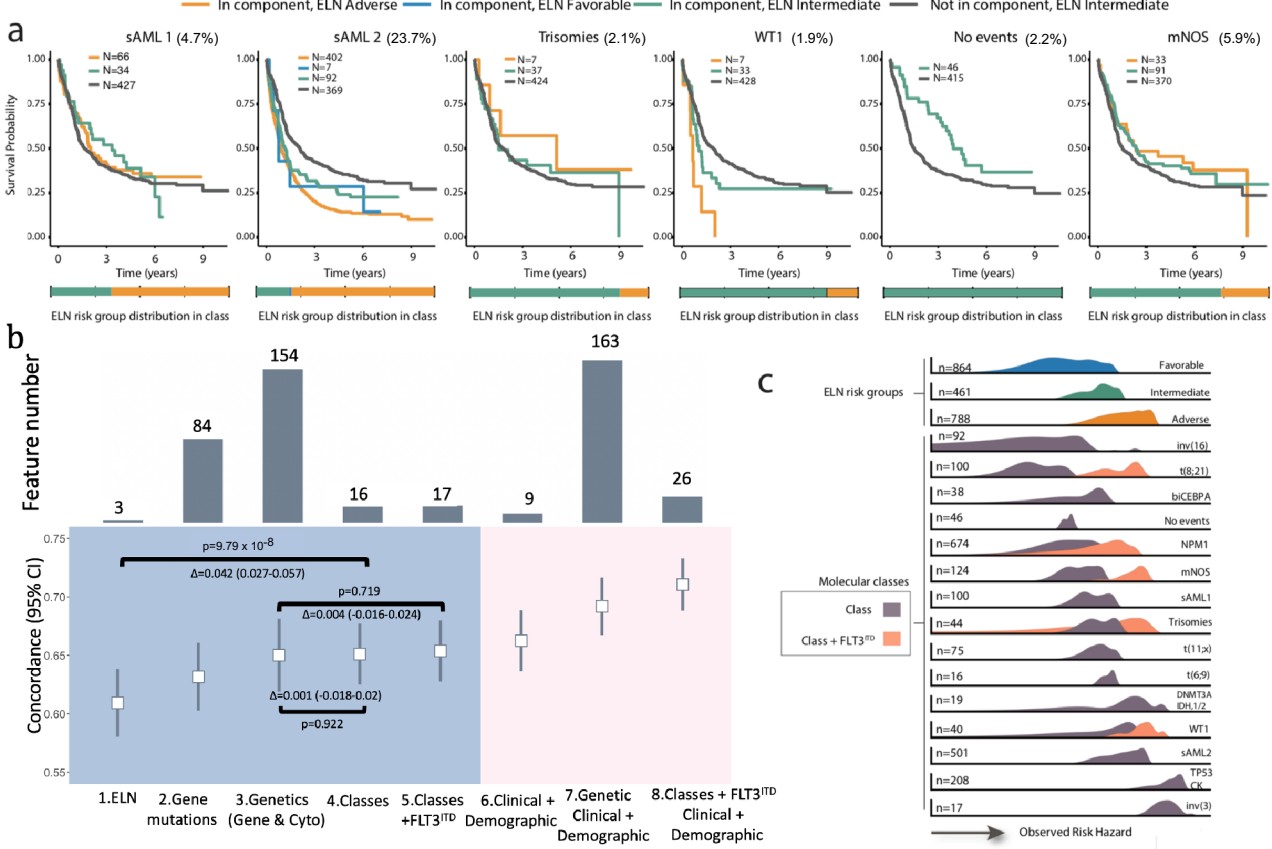

**Fig. 2 | Prognostic relevance of molecular subgroups. a** Kaplan–Meier overall survival curves for the sAML1, sAML2, trisomies, WT1, no event and mNOS subgroups, separated by ELN[2017] scores. A bar plot representing ELN[2017] repartition for each class is included in the lower panel. **b** Estimates of the concordance index (C-index) derived from Cox regression with a ridge penalty that consider (1) ELN[2017] strata, (2) gene mutations, (3) molecular classes, (4) molecular classes + *FLT3*[ITD], (5) genetic data (gene mutations and cytogenetics), (6) clinical and demographic, (7) genetic, clinical and demographic and (8) classes, *FLT3*[ITD], clinical and demographic features using internal 5 fold cross-validation for penalty selection. Top panel includes barplots representing the number of features/categories considered in

each model (i.e. 3 for ELN). The centers of the error bars represent the mean; the lower and upper whiskers represent the 95% CIs. Annotated *P*-values are from two-sided *t* score test. **c** Density plots representing the scaled observed hazard (0–1) for the ELN2017 risk categories and the proposed molecular classes. In purple we show the density of risk for each class, in orange we present the subset of cases in class that also have *FLT3*[ITD]. We omitted the density plot for class t(15;17) due to small numbers. The hazard is depicted for overall survival. In all boxplots, the median is indicated by the horizontal line and the first and third quartiles by the box edges. The lower and upper whiskers extend from the hinges to the smallest and largest values, respectively, no further than 1.5 × interquartile range from the hinges.

not "class defining" and are therefore not considered in the hierarchical classification schema.

### Integration of AML classes into prognostic models for clinical management

Prompted by the strong associations between class and outcomes, we compared prognostic models that considered genetic features to class-based models (Supplementary Tables 3, 4, Methods) or both, using ELN[2017] as a reference. Monosomal karyotypes[19] did not provide independent prognostic value[17] (Supplementary Fig. 21). Model comparison demonstrates that a simple model, based on class membership

and *FLT3*[ITD] status (17 features), captures the same prognostic information as more complex genetic models (154 features)(Fig. 2b, Supplementary Figs. 22–29). These findings provide a rationale for the development of a risk stratification schema that is based on class membership and *FLT3*[ITD] status, thus offering the opportunity to unify classification to risk stratification and importantly link a biological definition of disease ontology to clinical presentation and outcomes.

In agreement with prior findings[7], inclusion of clinical features (age of diagnosis, gender blast, antecedent hematologic disorder, performance status, white blood cells, hemoglobin and platelet) achieved the highest improvement in model discrimination (Fig. 2b,

Supplementary Fig. 30). Figure 2c exemplifies how the heterogeneity in clinical outcomes (as a function of overall survival hazard) is captured by the proposed classification. Despite differences in age, geography and chronology, feature selection was comparable in the AMLSG cohort, indicating that results are generalizable across AML patients (Supplementary Figs. 31, 32) and further demonstrating class membership as stable features for prognostic model construction in AML.

## A multi-state model for disease progression

We next studied associations between class membership, treatment response and relapse. Modeling a patient's journey through treatment, we applied a six-state Markov Model (MM)[20] that includes the following states: alive ($n = 2017$); alive in CR ($n = 1460$); relapse ($n = 778$); death without CR ($n = 543$); death with CR ($n = 199$) and death following relapse ($n = 607$) (Fig. 3a, Supplementary Fig. 33). Results were consistent in the intensively treated subset ($n = 1661$) (Supplementary Fig. 34).

This provides a detailed analysis of the proportion of patients in each class likely to transition between any two clinical endpoints over time (e.g Alive in CR -> Death in CR, or Alive in CR -> Relapse ->Death in relapse). The resulting survival estimates reflect the cumulative hazard for each of the transitions (Fig. 3b–d). This provides a detailed resolution of anticipated transitions across molecular subgroups. For example, patients with inv(16) or t(8;21) have comparable OS estimates, yet patients with inv(16) are more likely to relapse[21] (Supplementary Figs. 35, 36). Notably, upon relapse, inv(16) patients achieve the highest salvage frequencies, as compared to all other AML classes. Patients with no events, considered as intermediate-risk, have similar transitions to the *NPM1* class. We estimate endpoint-specific outcomes for the *WT1*, Trisomy and mNOS classes, which together with t(6;9) respond well to induction chemotherapy. However, patients in *WT1* and t(6;9) class have a high likelihood of relapse-related mortality (Fig. 3b–d). This is particularly the case for the subset of patients with $FLT3^{ITD}$. Indeed, subjects with $FLT3^{ITD}$ had both decreased likelihood of achieving CR, and increased risk of relapse-related mortality across all AML classes, not just *NPM1* (Supplementary Figs. 35, 36). This is despite the use of escalated doses of daunorubicin in AML17 which has been reported to reduce relapse risk in patients with FLT3$^{ITD}$ [22]. Furthermore, this model demonstrates that a key differentiator between sAML1 and sAML2 is response to induction chemotherapy, with 43.7% of sAML2 group patients not attaining CR as compared to 26% in sAML1 (Fisher's Exact test $p = 0.002$). Consistent with prior findings, adverse outcomes in TP53/complex and inv(3) are explained by highly chemoresistant disease and relapse-related mortality[14,23]. These observations were also observed in the AMLSG cohort (Supplementary Fig. 37, Supplementary Data 5).

## Implications for disease surveillance

MRD surveillance assesses initial response and guides treatment decisions[24,25], such as HSCT. Whilst MRD status is considered an independent predictor of outcome[26,27], the predictive relevance of MRD has not been determined across classes.

Results from MRD surveillance by flow-cytometry after course 1 were available in 523 UK-NCRI AML17 patients[16]. Of these, 202 were CR MRD$^{-ve}$ and 321 were CR MRD$^{+ve}$ (Fig. 4a, Supplementary Fig. 38). The MRD$^{+ve}$ rate, by class, ranged from ~33% to 95% (Fig. 4b). As expected[6], MRD$^{+ve}$ patients had a higher risk of relapse and death (Fig. 4a), with some exceptions. 70% (MRD$^{+ve}$ = 69) of sAML2 patients in CR were MRD + , yet while there was no evidence of a significant difference in relapse or survival rates, there was no difference of effect by group ($p = 0.3$ for interaction) (Fig. 4c, Supplementary Fig. 39). For sAML1, there was no difference in relapse-incidence between MRD$^{+ve}$ and MRD$^{-ve}$ subjects. A trend towards

poorer OS for MRD$^{+ve}$ patients was observed. These results suggest that while achieving MRD-negativity after the first course is associated with favorable outcomes, its utility may not be universal across classes (Supplementary Figs. 40–43) and that differences may be explained by the underlying biology associated with the mutations in each class.

## Relevance of AML classes to transplant outcomes

Next, we evaluated HSCT outcomes by AML class. Consolidating data from 2,244 intensively treated patients in the UK-NCRI ($n = 1095$) and AMLSG (total $n = 1149$) that achieved CR, 759 patients were transplanted in CR1 and 436 after relapse (Total $n = 1195$) (Supplementary Tables 5–7).

Evaluation of OS with respect to class and HSCT timing (Fig. 4d, e) demonstrated that sAML2 patients undergoing HSCT had a reduced risk of death, following adjustment for performance score and age ($p < 0.0001$; Fig. 4d). There was no significant survival difference based on HSCT in CR1 or CR2 ($p = 0.21$; Fig. 4e). Of note, patients in the TP53-CK also benefited from HSCT. However, in this group, patients transplanted in CR2 had significantly worse survival than those transplanted in CR1 ($p = 0.009$; Fig. 4e, Supplementary Fig. 44). Adjusting for age and performance score, HSCT was not associated with a reduced risk of death for patients in the sAML1 group, albeit there was evidence of a benefit for patients transplanted in CR1 vs CR2 (Fig. 4d, e, Supplementary Fig. 44).

Prior studies show that adverse-risk groups as defined by cytogenetics benefit from transplant[28]. Here our findings extend the definition of adverse-risk groups to include the newly defined sAML2 group, which account for 23.7% of patients in the study. However, given inherent selection biases associated with transplant, these results warrant validation in prospective studies.

## Relevance of molecular class in AML risk stratification

Using a panel of 32 genes (Supplementary Table 8), the proposed classification explicitly assigns 92% of patients into one of 14 AML classes and is sufficient to classify remaining patients into the two mNOS or no events subgroups. We demonstrated that class membership and $FLT3^{ITD}$ status capture the same prognostic information as genetic parameters. We next assessed how a class-based framework might inform future ELN$^{2017}$ revisions.

Using the ELN$^{2017}$ as a foundation, we assigned each class to one of three proposed risk strata (Favorable$^p$, Intermediate$^p$, Adverse$^p$) (Fig. 5a). Patients in the no events class were assigned to Favorable$^p$ group and "mNOS" patients to the Intermediate$^p$. sAML1, trisomies, *WT1*, *DNMT3A/IDH* and t(6;9) were classified as Intermediate$^p$-risk and sAML2 as Adverse$^p$. Per ELN$^{2017}$, t(6;9) was considered an adverse-risk group. We show that adverse-risk is specific to the subset of patients (72%) with $FLT3^{ITD}$ (Supplementary Fig. 45). We demonstrate that $FLT3^{ITD}$ is the only gene that delivers independent prognostic information from class membership.,Indeed, the presence of $FLT3^{ITD}$ was associated with worse outcomes for all intermediate-risk (Fig. 5b, Supplementary Fig. 46) classes. This association was independently of $FLT3^{ITD}$ ratio[29,30] (Supplementary Fig. 47). Thus, $FLT3^{ITD}$ status was used to upgrade risk for all intermediate-risk patients to adverse-risk.

Taken together, this framework re-stratified 25.5% of NCRI and 24.6% of AMLSG patients (Supplementary Figs. 48, 49). The redefined risk-strata overlapped with ELN$^{2017}$ trajectories. However, the proposed framework led to an increase in variance explained and a significant improvement in the c-index ($p = 0.05$ for NCRI; $p = 0.025$ in AMLSG) (Fig. 5c, Supplementary Fig. 50, Supplementary Table 4). The relative proportion of transplanted patients did not differ amongst the respective ELN strata (Supplementary Table 7) and results were consistent in the intensively treated subset ($n = 1755$) (Supplementary Fig. 51).

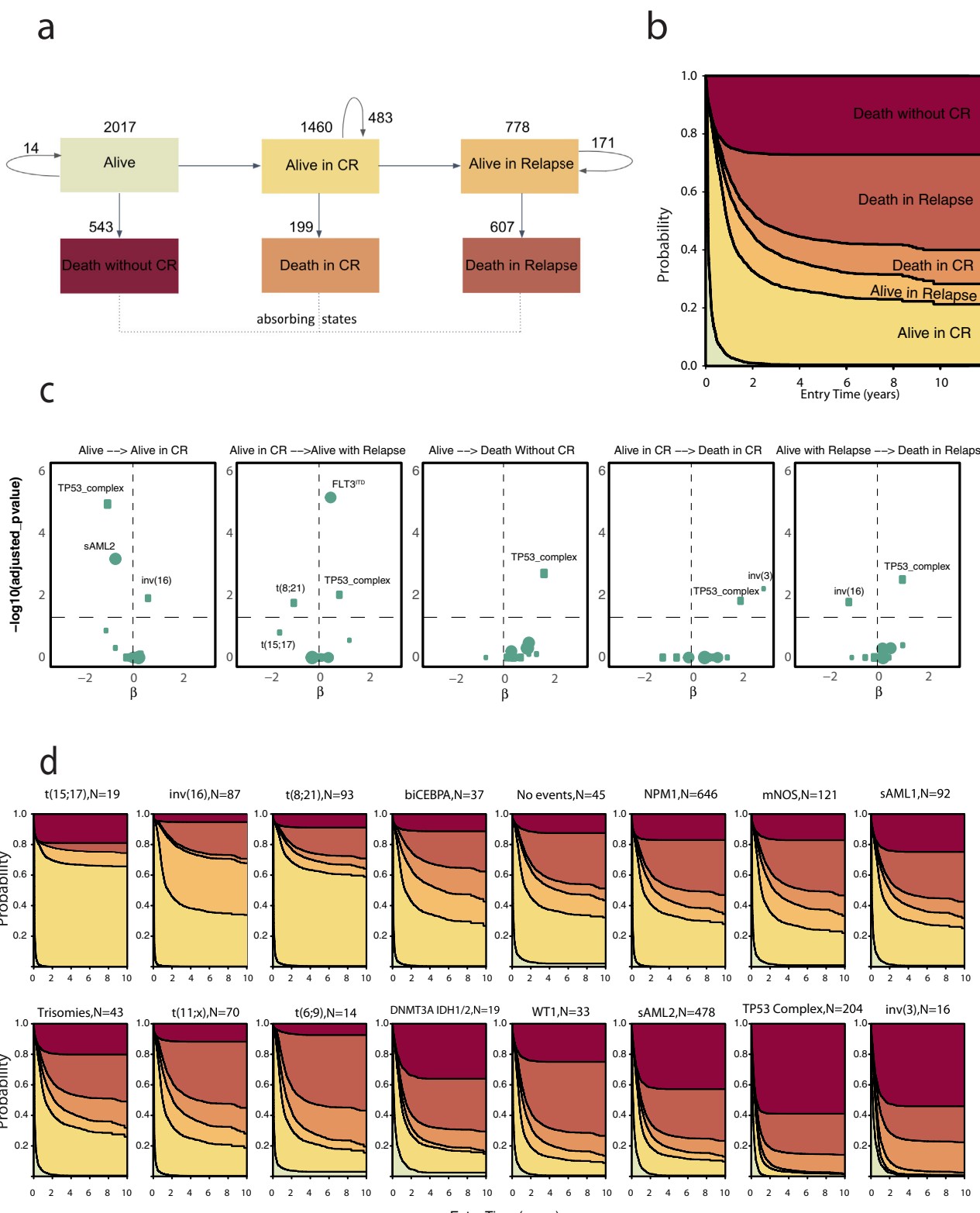

**Fig. 3 | Multi-state model for disease progression in the AML NCRI Cohort (n = 2017). a** Representation of patient transitions (in numbers) across clinical endpoints (alive (meaning received induction chemotherapy); alive in complete remission; alive in relapse; death without complete remission; death in complete remission; death in relapse).The arrows represent the number of transitioning patients. Circle arrows correspond to number of patients that do not transition. **b** Stacked transition probabilities (*y*-axis) across time (*x*-axis). **c** Cox volcano plots depicting the association between state transitions and molecular classes and *FLT3*[ITD]. The horizontal dotted curve corresponds to the *p*-value threshold of 0.05 and the vertical one corresponds to β = 0 on the *x*-axis. Highlighted predictors have a significant effect or have large β coefficients (*p*-value greater than the threshold: 0.05 here or *p*-value close to threshold and |β| >1.5). The size of each point corresponds to the frequency of the event. The reference class in the Cox transition models is no events. Wald test *p*-values are adjusted to correct for multiple comparisons. **d** Stacked transition probabilities for each class (*y*-axis) across time (*x*-axis). The bold lines represent the transition states. We omitted *n* = 96 patients from the multi-state model for disease progression (2113–96 = 2017) due to missing timepoints.

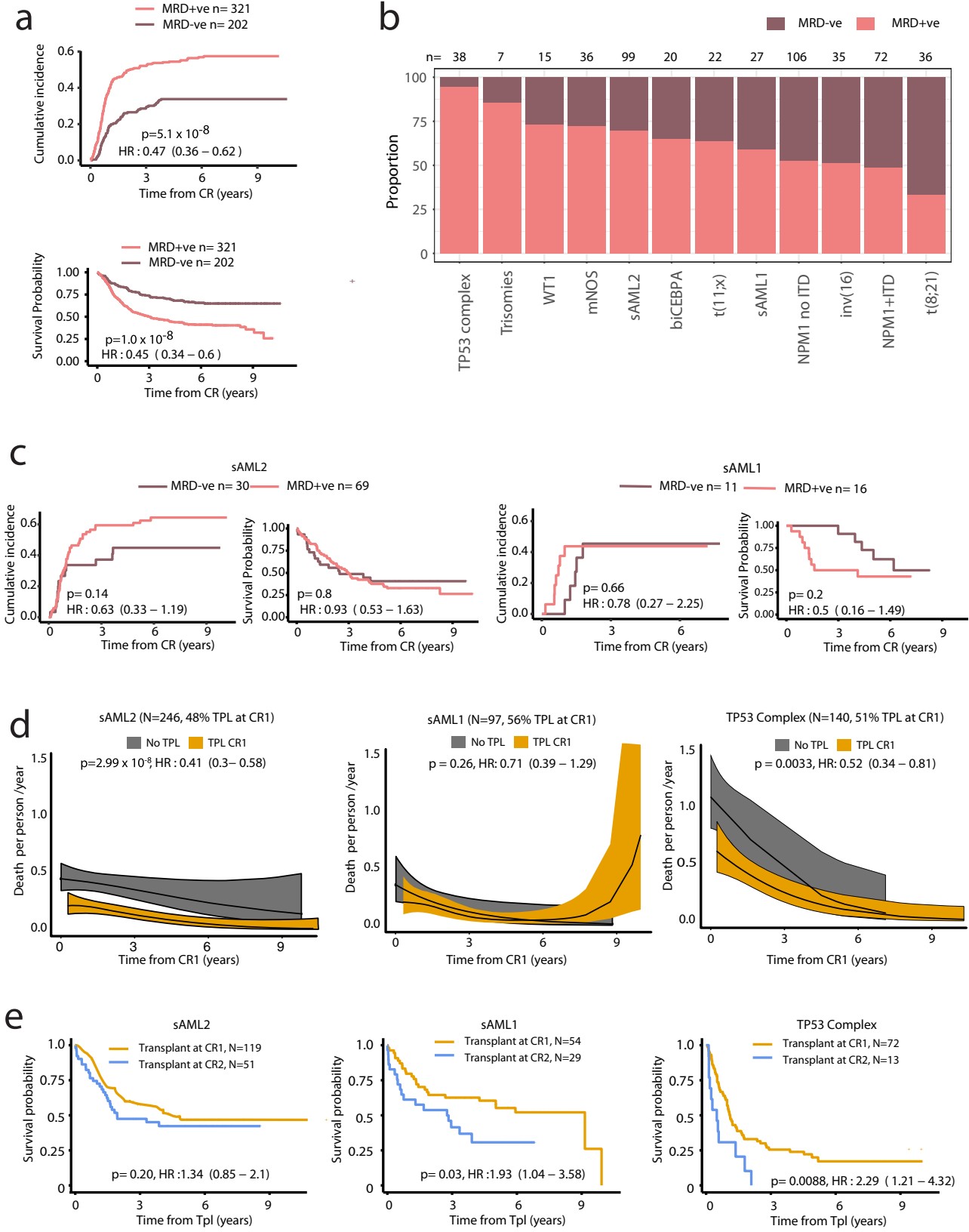

## Clinical decision support tools

Appreciating the complexity introduced by the multitude of genetic features considered, we developed a web-based tool that executes the proposed classification and risk stratification hierarchy (Figs. 1e, 5a) (Figs. 5, 6, https://www.aml-risk-model.com/calculator). Using mutations in 32 genes and cytogenetics as input variables, supervised

classification assigns each patient into the corresponding AML class and risk group. The model is restricted to the intensively treated subset of the study ($n = 3201$), which represents 90% of the patients in both NCRI and AMLSG cohorts. Graphical representation of end-point specific predictions across time are presented in the form of sediment and barplots. The contributing factor tab displays patient specific

**Fig. 4 | Implications for measurable residual disease surveillance and transplant outcomes. a** Cumulative incidence of relapse and Kaplan–Meier overall survival curves for patients that attained CR in AML17 trial subset, stratified by MRD status post course 1 ($n = 523$). Two-sided Gray's test and the logrank test were used to compare the relapse incidence and survival, respectively. **b** Barplots indicating proportion of patients in each molecular class with flow MRD +ve (any detectable MRD) or MRD−ve status post course 1. Restricted to the AML17 trial subset ($n = 523$) and to classes with at least five patients in the MRD + ve subset. **c** Incidence of relapse and OS by MRD status for the sAML2, sAML1 subgroups. A test for interaction between sAML1 vs sAML2 and MRD (Interaction HR: 1.90 (0.55–6.49), $p$-value: 0.31) was not significant. The analysis provided in **c** is limited to AML17 patients with MRD data available. Two-sided Gray's test and the logrank test were used to compare the relapse incidence and survival, respectively. **d** Nonparametric

estimated curves of the hazard rate (deaths per person-year; $y$-axis) across time ($x$-axis) for the sAML2, sAML1 and TP53 complex subgroups in the combined dataset (UK-NCRI and AMLSG). Curves display the hazard for patients transplanted (TPL) in CR1 to the non-transplanted patients. Tests of association were modeling transplant as a time-dependent covariate adjusted for age and performance status. A test for interaction between sAML1 vs sAML2 and transplant was borderline significant (Interaction HR: 0.57 (0.30–1.08), $p$-value: 0.08). 95% CIs are shown in the shaded areas. **e** Kaplan–Meier overall survival curves comparing patients who have been transplanted in CR1 to patients transplanted in CR2 for the selected classes. $P$-values are computed using the log-rank test. The analysis in **d**, **e** is limited to the patients to 2244 intensively treated patients in the UK-NCRI ($n = 1095$) and AMLSG (total $n = 1149$) that achieved CR, 759 patients were transplanted in CR1 and 436 after relapse (Total $n = 1195$).

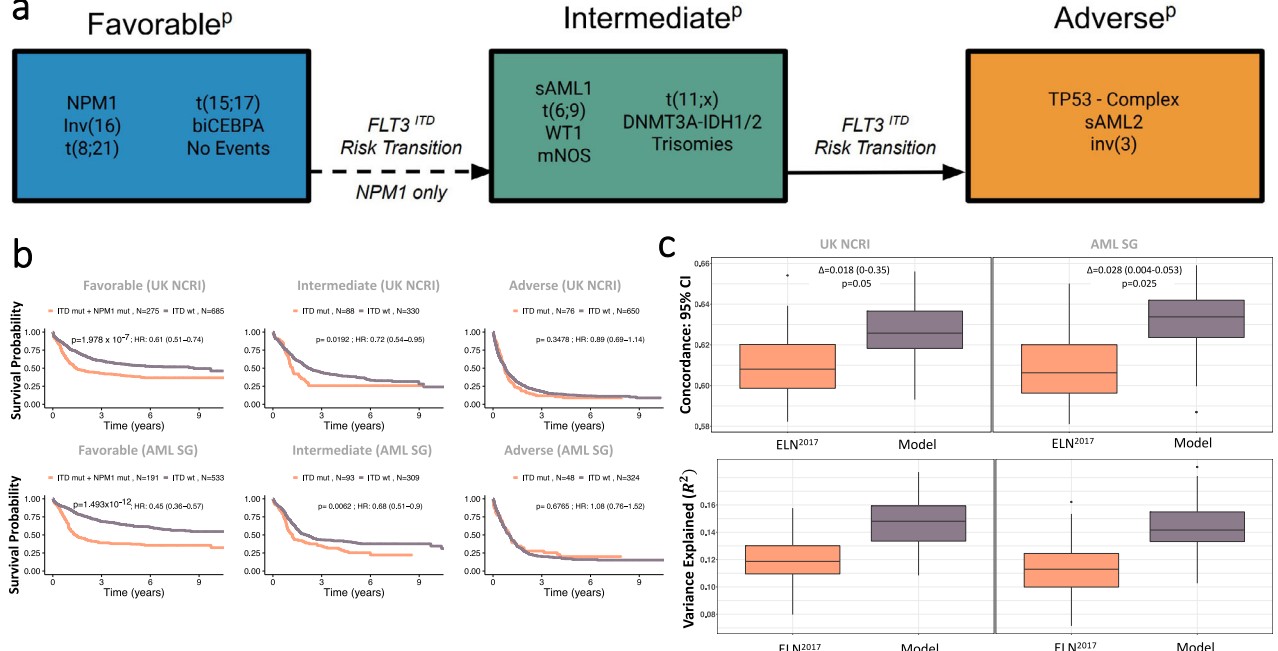

**Fig. 5 | Establishment of a new risk proposal based on the AML classes. a** Class assignment into one of three proposed risk categories (Favorable[P], Intermediate[P], Adverse[P]) is based on class membership and $FLT3^{ITD}$ status, whereby the presence of *NPM1* and $FLT3^{ITD}$ in the Favorable[P], the presence of $FLT3^{ITD}$ in the Intermediate[P] groups shifts one risk category to the Intermediate[P] and Adverse[P] respectively. *NPM1* $FLT3^{ITD}$ Patients classified as intermediate by class membership with the presence of $FLT3^{ITD}$ shift to adverse. The dotted arrow refers to the risk transition for patients with both *NPM1* and $FLT3^{ITD}$ mutations from favorable to intermediate. The solid arrow refers to the risk transition for patients with $FLT3^{ITD}$ from intermediate to adverse. **b** Kaplan–Meier overall survival curves comparing each of the proposed risk strata (Favorable[P], Intermediate[P], Adverse[P]) by the presence of *NPM1* and $FLT3^{ITD}$ status for the

Favorable[P] and by $FLT3^{ITD}$ status for the Intermediate[P] and Adverse[P] in the training AML NCRI cohort ($n = 2113$) and the validation AML SG cohort ($n = 1540$) validate the rationale for the $FLT3^{ITD}$ shift in risk. Annotated $P$-values are from two-sided log-rank tests. **c** The estimated improvement in the concordance index (C-index) and pseudo-variance explained (R2) for the two classifiers in the training AML NCRI Cohort ($n = 2113$) and validation AML SG Cohort ($n = 1540$). 95% confidence intervals were generated by bootstrap resampling for the C-index. In all boxplots, the median is indicated by the horizontal line and the first and third quartiles by the box edges. The lower and upper whiskers extend from the hinges to the smallest and largest values, respectively, no further than 1.5×interquartile range from the hinges. Annotated $P$-values are from two-sided t score test.

covariates that inform each transition estimate alongside the corresponding coefficients. For example, Patient PD25176a, classified as intermediate risk per ELN[2017] is 63 with normal karyotype and mutations in *BCOR* and *SF3B1*. Here, this patient classifies as sAML2 with Adverse[P] risk and the predicted outcomes for each transition are displayed (Fig. 6, Supplementary Fig. 52). To account for cases with clinical presentation outside the 95th quantile range of the training cohort and enhance interpretability of results, we introduce a warning sign for outlier cases and compute confidence intervals for all predictions in the calculator.

## Discussion

The scale and comprehensive analyses deployed in this study enabled us to validate some of the findings in the prior literature, establish

further insights and consolidate these into a global framework for the introduction of molecular biomarkers in clinical algorithms for AML patient management.

Using data from 3653 patients, we develop and validate a unified framework for disease classification and risk-stratification in AML that is informed by cytogenetics and 32 genes. This framework classifies 100% of AML patients into one of 16 molecular subgroups and refines our understanding of established classes (e.g t(6;9)), as well as provisional WHO entities (e.g *RUNX1*). We identify novel clusters of prognostic relevance (sAML1, sAML2, WT1, trisomies) accounting for 33.3% of AML patients, demonstrate the importance of negative molecular findings (No events, mNOS) and highlight the broad implications of $FLT3^{ITD}$-positivity irrespective of $FLT3^{ITD}$ allelic ratio.

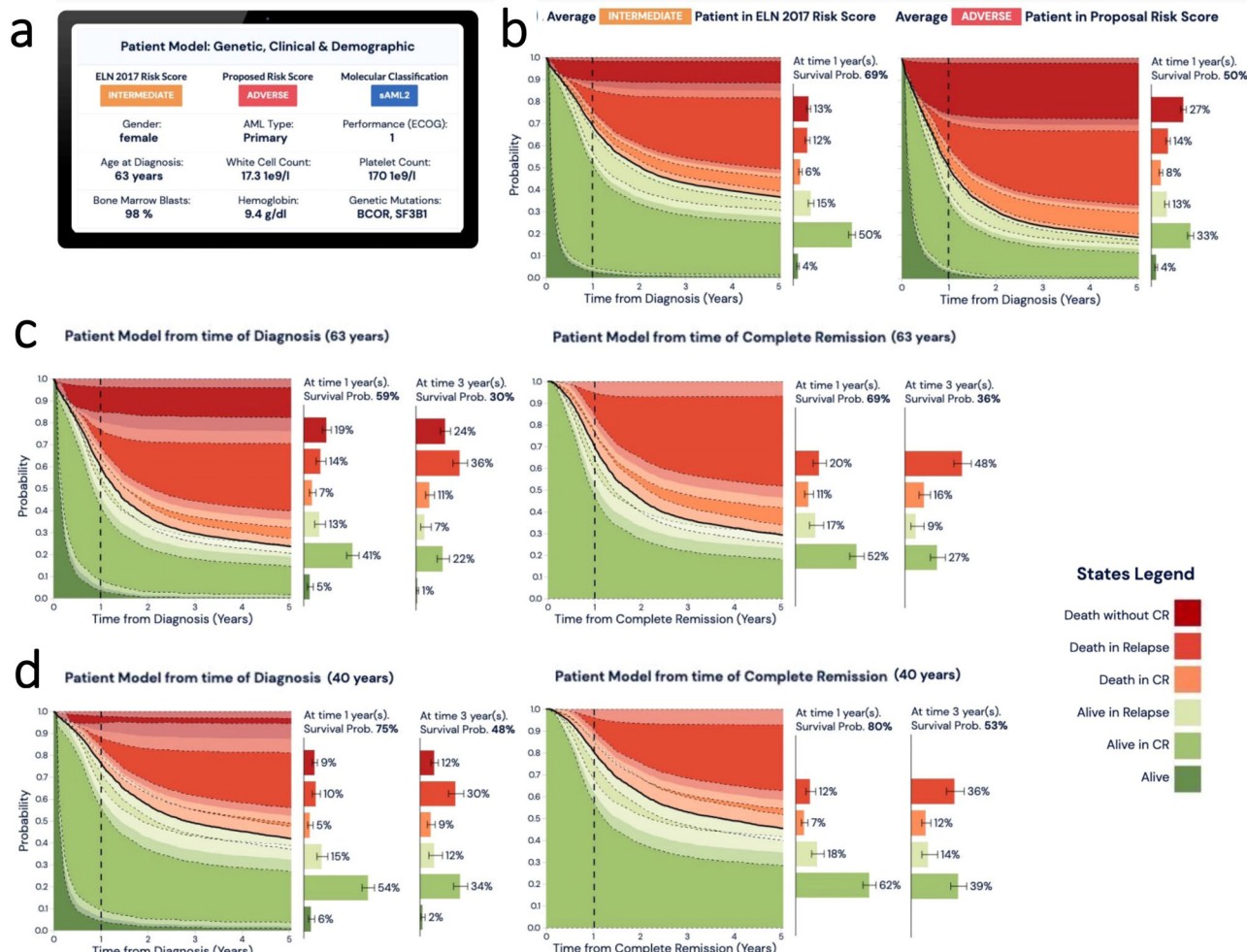

**Fig. 6 | Example presentation of personalized clinical decision support tool for molecular classification and risk stratification.** The calculator is derived using the multi-state models that consider data from ($n = 3201$ total patients, UK-NCRI and AMLSG) all intensively treated. **a** Input parameters to include cytogenetic, genetic, clinical and demographic are considered to (**b**) display each patient's ELN[2017] score alongside with the proposed risk group developed in this study. To further improve interpretation confidence, we provided confidence intervals for each sediment plot and probability estimate transitions. The lower and upper whiskers represent the 95% CIs. Stacked sediment plots for each patient represent the likelihood to transition between clinical endpoints across time. *X*-axis indicates time from diagnosis in years. *Y*-axis indicates probability for each transition. Black line indicates probability of survival across time. Vertical dotted line indicates the 1 year time frame. Horizontal barplots indicate the probability of attaining each one of these endpoints at selected time (vertical dotted line). **c**, **d** Adjacent barplots show the relative contribution of each covariate (molecular, clinical, demographic) on each transition. Estimates can be dynamically derived for the time of diagnosis or upon attainment of Complete Remission (CR) and across timepoints (i.e. Year 1 post diagnosis or CR, Year 3 post diagnosis or CR etc). To further improve interpretation confidence, we provided confidence intervals for each sediment plot and probability estimate transitions. The lower and upper whiskers represent the 95% CIs. Stacked sediment plots for each patient represent the likelihood to transition between clinical endpoints across time. *X*-axis indicates time from diagnosis in years. *Y*-axis indicates probability for each transition. Black line indicates probability of survival across time. Vertical dotted line indicates the 1 year time frame. Horizontal barplots indicate the probability of obtaining each one of these endpoints at selected time (vertical dotted line) and at 3 years.

Implementation of multistate models that consider each transition during a patients journey through AML, such as attainment of CR, likelihood of relapse and risk of death fine-map the most likely temporal trajectories for the established classes, and importantly further dissect and add granularity to the newly characterized classes in this study, which were previously merged into a heterogeneous unknown or intermediate-risk group. This provides a blueprint linking the biological processes deregulated within each class to a patient's likelihood of response to treatment, risk of disease progression, relapse and death. As an exemplar of this added clarity, the sAML2 class accounts for 24% of AML patients, is associated with chemorefractory disease, high relapse rates and poor survival, irrespective of early MRD negativity. However, for the subset of sAML2 patients who achieve CR, there appears to be a benefit of HSCT. Future studies, powered by adequate sample size are warranted to confirm these observations.

Building upon the ELN[2017] guidelines, we propose a three-tier risk-score (Favorable[P], Intermediate[P], Adverse[P]) informed by class membership and *FLT3*[ITD] status. Informed by the AML classification, this framework restratifies one in four AML patients and achieves significant improvement in prognostic accuracy. Moreover, despite demographic and clinical trial differences between our test and validation cohorts, our findings are reproducible across both the UK-NCRI and AMLSG cohorts. This is likely because the molecularly defined classes capture the spectrum of phenotypic and clinical heterogeneity observed amongst AML patients. Importantly, this demonstrates that findings from this study are generalizable across AML patients and are representative of those seen in routine clinical practice, particularly those considered fit for therapeutic intervention, where clinical decision making is currently most problematic. Despite the emergence of adjunct therapeutic approaches in the management of AML[3,4], the

backbone regiments in our training and validation cohorts ("7 + 3 like") are representative of AML treatment practices globally.

Recent correlative studies between molecular biomarkers and clinical outcomes primarily focus on single-genes or broad, heterogeneous risk groups[13,31–33]. This has challenged integration of findings into generalizable clinical algorithms to guide patient management. More recently, we prototyped patient-tailored clinical decision support tools in AML that deliver personalized risk scores[7]. However, in the absence of adjunct risk strata the utility of personalized risk scores in clinical trial design and interpretation of results is limited. Additionally, the complexity and "black box" nature of the models challenge clinical implementation and interpretation. Here, we deliver a simplified framework whereby mutations and cytogenetic findings at >100 loci can be summarized by 16 classes and the corresponding three risk strata. Class membership provides resolution in the heterogeneity observed in clinical presentation and delivers a rationalized schema for correlative studies as compared to single biomarkers or clinical cutoffs (e.g. % blast counts), which may dichotomize or group together heterogeneous and biologically distinct nosological entities. The delivery of a stable, biologically informed classification system further enables future studies of class-specific associations that extend beyond those of single gene-mutations and capture the most common patterns of co-mutation.

Integration of data from MRD and HSCT outcomes allowed us to prototype such correlative analyses using class-based associations. Indeed, the applicability of this framework to emerging treatment approaches in AML management to include less intensive regimens[4] or emerging therapeutics[3] needs further definition. Likewise, while we demonstrate a significant increase in risk hazard for many molecular classes in the presence of an additional $FLT3^{ITD}$ - mutation, we note that the survival advantage described in recent trials where $FLT3$ inhibitors were combined with intensive chemotherapy were modest[34–36]. It is noteworthy that patients with established adverse cytogenetics or genetic features ($TP53$ mutations) associated with poor outcomes to intensive chemotherapy also have adverse outcomes in response to emerging treatments such as azacytidine and venetoclax[4]. Here we expand the definition of adverse risk to encompass patients with sAML2 and a broader cohort of patients with $FLT3^{ITD}$ mutations.

As clinical management options evolve, studies focused on class associations with MRD status[19], response to emerging therapeutics[3] such as $FLT3$ or $IDH^{37,38}$ as well as combination regimens (e.g azacytidine and venetoclax)[4] are warranted to make definitive determinations for AML patient care.

## Method

Informed consent was obtained for all subjects to the study.

All relevant ethical guidelines have been followed, and any necessary IRB and/or ethics committee approvals have been obtained. The trial was conducted in accordance with the Declaration of Helsinki, was sponsored by Cardiff University, and was approved by the Wales Research Ethic Committee. Sample collection was approved by the Wales research ethic committee protocol number 08/MRE09/29 Analysis of the data in this study was approved by MSKCC Institutional Review Board protocol number x20-064.

### Clinical annotation

Demographic and diagnostic variables were ascertained at the time of diagnosis by the NCRI Clinical Trial Center. The variables collected included: Age of diagnosis (AOD), Gender, Bone marrow karyotype, AML type (AML: de novo AML, sAML: secondary AML, tAML: therapy related AML), Antecedent hematological disease (AHD), Performance status and peripheral blood counts to include white blood cell (WBC) counts, lactate dehydrogenase levels (LDH), platelet counts, hemoglobin levels (Hb). Outcome endpoints included status at last follow up, time from diagnosis to time of last follow up, event free survival

and time to event, relapse free survival and time to relapse as well as whether complete remission was achieved, time to remission and time to transplant where transplant was received.

For patients in the NCRI AML17 trial flow cytometric MRD data were previously derived from Freeman et al.[27]

### Molecular annotation

**Preparation of custom capture libraries for sequencing.** Genomic DNA was extracted from peripheral blood or bone marrow mononuclear cells for adult AML patients ascertained through the UK-NCRI AML Trials AML11, AML12, AML14, AML15, AML16, AML17 and AML Li1 (Supplementary Table 1). All samples were collected at diagnosis prior to any treatment. Custom RNA baits were designed per manufacturers' guidelines (SureSelect, Agilent, UK) complementary to all coding exons of 128 genes as well as genome wide SNP probes selected using the following criteria:(1) Inter-marker distance every 3 MB; (2) Population Minor Allele Frequency > 0.4; (3) Not within repeat regions. Detailed bait file design to include all regions targeted by RNA baits, alongside genomic coordinates and annotation are provided in Supplementary Data 1.

A total of 125 μl of 40 ng/μl of WGA DNA was fragmented to an average insert size of 145 bp (75I300) and subjected to Illumina DNA sequencing library preparation using the Bravo automated liquid handling platform. Individual samples were indexed using a unique DNA barcode via six cycles of PCR. Equimolar pools of 16 libraries were prepared and hybridized to custom RNA baits following the Agilent SureSelect protocol. Enriched pools of 96 cases were sequenced on a lane of an Illumina HiSeq Genome Analyzer machine using the 75 base pair paired-end protocol up to a median coverage of 600x. Only data that passed FASTQC MultiQC criteria (coverage > 100x, duplicate reads, GC bias) were retained in the study.

Of note, all samples from the NCRI trials were uniformly processed, sequenced and annotated. AMLSG data were derived from the data repository in Papaemmanuil et al.[10].

All raw data has been deposited in the European Genome-Phenome Archive EGAS00001000570.

**Sequencing data alignment.** Raw sequence data were aligned to the human genome (NCBI build 37) using BWA[53]. Unmapped reads, PCR duplicates and reads mapping to regions outside of the target region (merged exonic regions + 10 bp either side of each exon) were excluded from analysis. Bedtools® coverage v2.15.05was subsequently used to determine the coverage depth at each base. Genes with median target coverage <20x were removed from the study and samples with median overall coverage <50x were also excluded from downstream analysis and are not reported in this study.

### Variant calling

**Substitutions.** Base substitutions were mapped using established bioinformatics approaches as previously described[10,54,55]. Single base, somatic substitutions were called independently in each sample using an in-house algorithm CaVEMan: Cancer Variants through Expectation Maximisation[56]. The algorithm compares sequence data from each tumor sample to an unrelated normal sample and calculates a mutation probability at each base-pair position locus. A number of post-processing filters were applied to improve specificity. Filters applied to targeted capture data required that:

1. At least a third of the alleles containing the mutant must have base quality ≥ 25.
2. If mutant allele coverage ≥ 10, there must be a mutant allele of at least base quality 20 in the middle 3rd of a read. If mutant allele coverage is < 10, a mutant allele of at least base quality 20 in the first 2/3 of a read is acceptable[10].
3. The mutation position is marked by <3 reads in any sample in the unmatched normal panel.

4. If the mean base quality is <20 then <96% of mutation-carrying reads are in one direction.

5. Variants were cross-referenced with a panel of unmatched normal samples ($n = 300$) to allow definition of base pair specific errors in the panel/

6. Previously reported bona fide somatic variants presenting in the unmatched normal panel were not filtered out from the dataset.

**Small insertion and deletions.** Small somatic insertions and deletions (indels) were identified using an in-house modified version of Pindel[57]. Post-processing filters were applied as previously described[58,59]. The following steps were taken to improve specificity for calling non-coding indels:

1. 'SUM-MS' score (sum of the mapping scores of the reads used as anchors) $\geq 200$

2. 'Previously Rejected Score' (PRS) is = =0

3. Bidirectional (evidence in both read directions (forward and reverse) in Pindel or BWA reads)

4. Variant allele is not a unit within a homopolymer track presenting with variant allele fraction <8%.

5. Variants did not present in ~300 unmatched normal samples and did not have a COSMIC ID with confirmed somatic status in the literature.

6. Artifactual indels occur at recurrent loci across multiple samples, often as a consequence of highly repetitive sequence. To ensure that such variants were not retained in the data, in-house databases for recurrently rejected Pindel calls interrogation were performed. All variants were visually inspected prior to removal.

Regions enriched for GC content and low target coverage were manually reviewed (i.e. *CEBPA, SRSF2*) and, where available, prior data derived by a CLIA approved diagnostic laboratory, were cross-referenced and for *FLT3*[ITD], *CEBPA* mutations and *NPM1* mutations incorporated in the dataset.

Additionally, for *FLT3*[ITD] detection a custom analysis script that performs a localized query for reads consistent with an inverted tandem duplication within the *FLT3* locus was developed.

Visual inspection using visualization software (IGV) was performed of all variants in the targeted gene screen dataset after applying these filters.

**Quality control and variant annotation.** To evaluate the relevance of each mutation in the study, we searched the following databases for the presence of each mutations and relevant annotation on clinical relevance: gnomAD https://gnomad.broadinstitute.org, COSMIC https://cancer.sanger.ac.uk/cosmic, cBioPortal for Cancer Genomics https://www.cbioportal.org, OncoKB Precision Oncology Knowledge Base https://www.oncokb.org, ClinVar https://www.ncbi.nlm.nih.gov/clinvar.

From the list of high confident somatic variants, putative oncogenic variants were distinguished from variants of unknown significance (VUS) based on:

- Recurrence in the Catalog Of Somatic Mutations in Cancer (COSMIC)[9], in myeloid disease samples registered in cBioPortal[60,61] or in the study dataset.
- The inferred consequence of a mutation; where nonsense mutations, splice site mutations and frameshift indels were considered oncogenic for likely tumor suppressor genes (from COSMIC Cancer Census Genes or OncoKB Cancer Gene List).
- Presence in pan-cancer hotspot databases[62,63].
- Annotation in the human variation database ClinVar[64].
- Annotation in the precision oncology knowledge database OncoKB[54].
- Recurrence with somatic presentation in a set of in-house data derived from >6,000 myeloid neoplasms[10,55,59].

Briefly using these database variant annotation parameters are listed by variant type as follows:

**Oncogenic**

- Known oncogenic variants previously reported in the literature or databases;
- Novel recurrent variants ($n \geq 2$) that cluster with known somatic variants in well characterized myeloid driver genes;
- Truncating variants (nonsense mutations, essential splice mutations or frameshift indels) in genes implicated in myeloid malignancies through acquisition of loss of function mutations;

Details of all mutations annotated as oncogenic and retained for bioinformatic analyses are provided in Supplementary Data 2.

**Variants of unknown significance.** Variants identified outside the range of frequent driver variants in genes with known oncogenic variants; Variants in genes whose role in myeloid disease is not yet established or variants presenting uniquely in the dataset were not considered in the study.

**Cytogenetic findings and copy number alterations (CNA).** Karyotypes for 983/2113, 46.5% were ascertained in accordance with the International System for Human Cytogenetic Nomenclature from diagnostic assessment. Recurrent alterations including fusion genes and copy number alterations were included in downstream analysis. Additionally we used Allele Specific Copy Number Analyses of Tumors (ASCAT[65]) to derive aberrant copy number segments at an arm or chromosome level resolution. Findings from CNA and cytogenetic findings from karyotype analyses were evaluated for concordance, as well as matched for patient gender, as a means to quality control that the derivative sample does indeed match the corresponding clinical data. Detail annotation of the recurrent copy number alterations and cytogenetic findings incorporated in the analyses is provided in Supplementary Data 3.

**Annotation of TP53 allelic state.** *TP53* allelic state[15] was annotated as follows:

- **Mono-allelic annotation** for samples with only one putative somatic mutation (substitution, small insertion or deletion, splice site mutation).
- **Multi-hit annotation** for samples that satisfied the following criteria:
    a. At least 2 mutations in TP53 (substitution, small insertion or deletion, splice site mutation).
    b. At least one mutation of TP53 and a concomitant deletion of the TP53 locus to include focal deletions, 17p deletions or whole chromosome 17 deletions.
    c. At least one mutation in TP53 at a VAF estimate >65%, indicative of LOH.

## Statistical methodology

**Bayesian dirichlet process for derivation of class membership.** We performed an ab initio evaluation of molecular classification in AML. We used a Bayesian Dirichlet process (BDP)(https://github.com/nicolaroberts/hdp), which defines a potentially infinite prior distribution for the number and proportions of clusters in a mixture model as previously described[10]. Briefly, using this process a dataframe of 2150 rows (for each patient) with 153 columns representing recurrent genetic alterations as binary variables (1 for present, 0 for wild type) is used as input (Supplementary Data 3-4). The optimal number of clusters is learned from the data by Markov chain Monte Carlo (MCMC) methods. This approach does not constrain genetic features within one cluster but rather allows genetic lesions to be shared across clusters. This means that the resulting clusters can share genetic

features. As the clusters are defined, patient samples obtain a probability of class membership for each cluster.

A 2-step Bayesian Dirichlet Process with Gaussian distribution was implemented. In the first iteration a total of x high-confidence clusters were defined assigning x patients with a high probability of assignment >x%. Once components are determined, for each patient, a probability of assignment to each of the components is derived on the basis of the representative features for each class. Hyperparameter selection was dependent upon: (1) The total number of high-confidence components; (2) Maximum probability of assignment for each patient; (3) The delta between the maximum probability of assignment and the second highest probability of assignment. A second iteration followed, with input the subset of patients that were not classified in the first iteration, which led to the x patients that did not classify within the first iteration.

Selection of the execution parameters (i.e. hyperparameter) selection was performed using a gridsearch (Supplementary Table 9).
- Cosine similarity threshold applied on the measure of distance between clusters to differentiate distinct components.
- Initial number of clusters: random initialization to this predefined number of clusters before applying the Gibbs sampling procedure.
- Base distribution: The distribution from which the parent node will draw from representing the shape of the data.
- Concentration Parameter $\alpha_A$: shape hyperparameter for the gamma prior over the concentration parameter.
- Concentration Parameter $\alpha_B$: rate hyperparameter for the gamma prior over the concentration parameter.

**Post processing.** High-confidence components were defined as non-overlapping clusters with uniform molecular composition that resulted in the highest probability of assignment of the constituent patients. Whilst each patient derives distinct probabilities of assignment for each cluster (i.e. Let Patient 1 with 60% assignment probability for cluster 1, 24% probability for cluster 2 and 16% probability for cluster 3), Patient 1 would be assigned to cluster (1) Patients were only assigned in a given component if the main class defining genes or cytogenetic alterations were present. For example, for a patient to be part of the NPM1 cluster, the patient has to have an *NPM1* mutation. Certain components, owing to similarity in patterns of commutation were assigned within the same cluster. These included t(15;17) and t(11;x), WT1 and t(6;9). A manual split into independent components was applied. For 91 patients in the study (4.3%) assignment criteria to >1 class post processing were fulfilled.

**Hierarchical class assignment.** An overall hierarchy for AML classification was informed by the presence of unique and non-overlapping molecular subgroups, class defining alteration frequency, class size. For overlap cases, size, clinical presentation and severity of clinical phenotype and outcomes was prioritized in the hierarchy (i.e. established WHO entities preceded entities, *TP53* and complex karyotype preceded the hierarchy a patient with *WT1* mutations).

Main Fig. 1e outlines the hierarchy of assignment in the proposed classification in an R snippet code

**Survival analysis.** Overall survival (OS) defined as the time to death or last follow-up was the primary endpoint of the study. Survival analyses was used by implementation of established statistical methodologies for modeling outcomes to include:

**Cox penalized models.** Cox regression model[39,40] with regularization using a Lasso penalty[41]. The degree of shrinkage λ was selected internally by cross validation. The Lasso penalty has the effect of forcing some of the coefficient estimates, with a minor contribution to the

model, to be exactly equal to zero. Cox Random Effects Models (https://github.com/mg14/CoxHD) were applied to introduce a Ridge type regularization shrinking the effect of correlated variables.

**Cox boosting $L^2$ models.** Cox likelihood-based boosting[42] approach with the negative $L^2$-norm penalized partial likelihood were performed.

**Random survival forests.** Random survival forest[43] models with the logrank splitting criteria were used.

**Support Vector Machine Survival (SVM).** SVM[44] with truncated Newton optimization and order statistic trees that did not rely on the number of comparable pairs of events was applied.

**Evaluation metric using the concordance index(C-Index).** The discriminaton of various models were compared using the concordance index (C-Index)[45]. The C-Index applied for all comparisons used an inverse probability of censoring weight (IPCW) to account for censored observations.

Models that consider different feature groups (genes mutations, cytogenetic findings, clinical variables, demographic variables, molecular classes and/or ELN risk strata) were evaluated across algorithmic approaches using the following workflow: 75% of patients in the dataset were randomly selected for the training and cross-validation. Model performances were evaluated on the 25% of remaining patients to compute the C-Index (evaluation metric).95% confidence intervals for the C-Index estimates were constructed using bootstrap resampling (100 bootstrap resamples). P-values were calculated using the C-Index estimates along with the standard error estimates from the bootstrap resampling. The same process was conducted for training and validation cohorts respectively (2113 patients for AML NCRI cohort and 1540 for AML SG cohort).

A number of algorithmic approaches were evaluated to include following hyperparameters selection:
- Cox model with penalization:
    - different values of $\alpha$: (0 to 1 with 0.2 increment) controlling the tradeoff Ridge-Lasso
    - internal validation of λ parameter controlling the weight given to the coefficients penalization
- Cox boost with the following hyperparameters:
    - maxstepno=500: maximal number of steps to evaluate
    - $K = 10$: number of folds to be used for cross-validation to find the optimal number of boosting steps
    - type = "verweij": used to compute the partial likelihood in the hold-out folds with Verweij method
    - penalty=100: penalty value for the update of an individual element of the parameter vector in each boosting step.
- Cox random effects with the following hyperparameters:
    - Groups: as many groups as number of variables and with
    - $v = 0$ meaning no hyperprior
    - $\sigma_0$: the variance of a si-chisq hyperprior on the variance
    - max.iter = 500: maximal number of iterations
    - tolerance = 0.01: the stopping criteria
- Random forest survival:
    - Nodesize: 5,10,20
    - Number of trees: 100–1200 with 50 increments
    - Split rule: log-rank splitting
    - tolerance = 0.01: the stopping criteria
- Support vector machine survival:
    - Kernels: linear, polynomial, radial basis function and sigmoid
    - $\alpha$: range 1e-6 to 1 multiplying by 10 to penalize the square hinge loss in the objective function. Internal cross validation was used to select this parameter.
    - Avltree optimizer named after inventors Adelson-Velsky and Landis

- ○ Max iter = 1000: maximal number of iterations
- ○ Tolerance = 1e-6

Details about softwares and packages used are available in Supplementary Data 6.

Algorithmic comparison on the best performing feature combination was also investigated in Supplementary Fig. 53.

**Feature importance.** A permutation strategy was developed to evaluate feature importance. The reference C-Index (evaluation metric) was computed and the values of each feature were permuted 50 times to compare the new C-Index with the reference C-Index without permutation. With this framework, each feature is ranked from the most to the least important for model performance using the values of the ratio = ref_CI / permuted_CI. This strategy is useful to characterize the relative association of each covariate with the given endpoint (i.e. survival) without information on direction of effect. This strategy was also compared with (1) the variable importance returned by random forest and (2) the coefficients given by the Cox penalized models and results were ranked similarly. This permutation strategy was applied for different feature combinations on the following models: Cox Lasso, Cox Ridge, Cox ElasticNet, Cox Random Effects and Random Forest Survival with the same parameters as presented in the above section.

**Multi-state models to model transitions across clinical endpoints.** Multi-state models[46,47] were developed to describe a stochastic process where patients experience a succession of events (alive (received induction chemotherapy), complete remission, relapse and death) and transition between different states overtime. Markov assumption is implicitly present in the likelihood of state transition: the future depends on history only through the present.

First, a non-parametric approach was performed to estimate overall transition probabilities within the cohort. Second, a semi-parametric approach was used by incorporating covariates of interest into the modeling of transition probabilities with different baseline transitions. The implementation to model those transitions was performed using the *mstate* package in R.

**Additional statistical tests.** Fisher's exact test[48] was used to compare patterns of co-mutations for each genetic abnormality. Due to multiple comparisons, all P-values were adjusted using the false discovery rate (Benjamin-Hochberg procedure). To compare continuous correlates (age,wbc,hb,plt and blasts) by mutational or class status, a Wilcoxon rank-sum test[49] was used. To compare other categorial correlates (gender,ahd,perf_status,secondary,eln) by mutational or class status, a chi-square or Fisher's exact test were conducted, as appropriate. A test of interaction within a Cox proportional hazards model was used to evaluate whether a giving class modifies the effect of MRD or transplant on survival or relapse. Lastly, survival was compared across two or more groups using a logrank test[50].

**Regression analysis for clinical variables.** Univariate regression volcano plots: Linear model was fitted using the *lm* package in R for continuous outcomes and a logistic regression (logistf package in R) for binary outcomes. In the volcano plot, each covariate coefficient on the x-axis and the log (base 10) false discovery rate adjusted p-value value on y-axis were plotted.

Multivariate regression coefficient rankings: Linear and logistic Lasso models were fitted using the *cv.glmnet* R package with internal cross validation of the regularization parameter λ. Each model used the entire dataset 100 times and the coefficients were aggregated for each of the covariates to evaluate absolute ranking.

**Hazard risk over time and risk density.** The *bshazard* package in R was applied to obtain a non-parametric smooth estimate of the hazard function based on B-splines[51].

In Fig. 2 of the manuscript, the hazard density across the ELN[2017] risk categories and the classes were plotted to visualize a shift in the hazard across the classes. The x-axis represents the hazard and the y-axis was rescaled to aid for the visualization of the hazards.

**Estimation of contributing factors in the multistate model**
We used the Cox semi-parametric approach to estimate the different transition probabilities where we specify covariates to be incorporated in the model. The transition hazard takes the following form:
- $h_{ij}(t \mid Z) = h_{ij,o}(t) \exp(\beta^T_{ij} Z_{ij})$, where $\beta^{(k)}_{ij}$ is the coefficient computed from the Cox transition model corresponding to covariate (k) for the transition from state i to state j, $Z_{ij}^{(k)}$ is the value for covariate (k) for that transition.

Thus, we compute the contributing factors for each transition:
Factor$^{CONTRIB}$ = $\beta_{ij}^*(Z_{ij} - Z_{ij,median})$ values, with $Z_{ij,median}$ the median value for that covariate in this transition.

A negative Factor$^{CONTRIB}$ value corresponds to a low risk covariate value transition while a positive value corresponds to a high risk covariate value transition.

We omitted n = 96 patients from the multi-state model for disease progression (2113 − 96 = 2017) due to missing timepoints.

**Development of study web portal**
**Data accessibility and code reproducibility.** All raw data have been deposited on EGAS00001000570. Additionally. a web portal to accompany this publication has been deployed on: https://www.aml-risk-model.com The portal includes (1) A direct link to cbio portal containing detailed molecular and clinical findings in the study; (2) List of gene panel used to develop the AML classification and risk stratification model; (3) A gene by gene description of all genotype-clinical, genotype-genotype, and genotype-outcome associations in the study; (4) A class by class description of all class-clinical, class-genotype and class-outcome associations; and (5) A personalized risk calculator tool. Additionally a Github[52] https://github.com/papaemmelab/Tazi_NatureC_AML has been deployed that contains all input data, code and data visualizations relating to this publication.

**AML risk calculator.** For the development of a personalized risk calculator we consider 3201 intensively treated patients from the AML NCRI and AMLSG cohorts. For each patient,cytogenetic findings and mutations in the panel of 32 genes were considered to derive class assignment, ELN[2017] and proposed risk scores.

Cox multi-state models[22,23] were used to estimate the transition covariate coefficients across six clinical endpoints (Alive (received induction chemotherapy) → Alive in CR; Alive → Death no CR; Alive in CR → Alive in Relapse; Alive in CR → Death in CR; Alive in Relapse → Death in Relapse) for the following 4 models: (1) AML Classes; (2) ELN[2017]; (3) Proposed risk score; (4) AML Classes, demographic and clinical parameters. For detailed information on the coefficients please refer to: https://www.aml-risk-model.com/calculator

For any patient outside of the cohort, given any combination of molecular, clinical and demographic parameters the web calculator determines a patient's class assignment, ELN[2017] and proposed risk score and estimates the transition probabilities for each of the six clinical endpoints. All of the 32 genes and cytogenetic findings are assumed to be wild type unless specified by the user. Clinical and demographic parameters that were not specified were imputed as the median for the cohort.

To increase confidence in the model interpretation, we have added the following features in the calculator:

1. Display of the overall cohort distribution for all clinical variables. This allows a practicing physician to appreciate whether the presentation of a given patient is within the expected or outlier ranges of the cohort.
2. Where input parameters for a patient represent outlier values, we have introduced a warning sign alerting the end user that the data supporting predictions for this patient may be less powered.
3. For the sediment plots, displaying probabilities of transitioning between each clinical point, we have added confidence intervals for the predictions. Confidence intervals are also available for the probability estimates.

A web application was developed, where the risk model is deployed as a serverless lambda function available through a restful API, and consumed by a single page javascript application. The web calculator is built using Python and Javascript open-source libraries for web development, and the R multi-state model is uploaded and executed on AWS Lambda in real-time to adjust for personalized transition probabilities prediction relative to the user's input parameters. Details of the services infrastructure and the cloud implementation using Amazon Web Services are shown in Supplementary Fig. 54.

## Cohort description
Mutation annotations are available for all AML NCRI and all AML SG patients (Supplementary Table 1).

AML NCRI cohort includes patients from AML17, AML16, AML11, AML12, AML14,AML15 and AML Li1. AML SG cohort includes patients from HD98A, HD98B and 07-04.

Transplant data is available for a subset of AML17 patients and a subset of AML SG patients (Supplementary Table 5).

MRD data is available for a subset of AML17 patients.

## Reporting summary
Further information on research design is available in the Nature Research Reporting Summary linked to this article.

## Data availability
Clinical, copy number and mutation data are available at https://github.com/papaemmelab/Tazi_NatureC_AML[52] and Supplementary Data 7. The data underlying all Main Figures, Supplementary Figures are provided as Supplementary Data 7 and available at https://github.com/papaemmelab/Tazi_NatureC_AML[52].

All raw data has been deposited in the European Genome-Phenome Archive EGAS00001000570.

## Code availability
Source code and data to reproduce all figures in the manuscript are available at https://github.com/papaemmelab/Tazi_NatureC_AML[52] and Supplementary Data 8.

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

## Acknowledgements

E.P. is a Josie Robertson Investigator and is supported by the European Hematology Association, American Society of Hematology, Gabrielle's Angels Foundation, V Foundation and The Geoffrey Beene Foundation and is a Damon Runyon Rachleff Innovator fellow. Work in the B.J.P.H.

lab is funded by Cancer Research UK (C18680/A25508), the European Research Council (647685), MRC (MR-R009708-1), the Kay Kendall Leukemia Fund (KKL1243), the Wellcome Trust (205254/Z/16/Z) and the Cancer Research UK Cambridge Major Centre (C49940/A25117). This research was supported by the NIHR Cambridge Biomedical Research Centre (BRC-1215-20014), and was funded in part, by the Wellcome Trust who supported the Wellcome—MRC Cambridge Stem Cell Institute (203151/Z/16/Z). The views expressed are those of the authors and not necessarily those of the NIHR or the Department of Health and Social Care. L.B., H.D. and B.J.P.H. are supported by the HARMONY Alliance (IMI Project No. 116026; https://www.harmony-alliance.eu/). The UK-NCRI AML working group trials were supported with research grants from the Medical Research Council (MRC), Cancer Research UK (CRUK), Blood Cancer UK and Cardiff University. We would like to thank all patients and investigators for their participation in the trials and the study.

## Author contributions

E.P., B.J.P.H., and S.M.D. designed the study. Y.T. performed all statistical analysis. S.M.D. and E.P. supervised statistical analysis plan and execution. E.P., B.J.P.H., S.M.D., and N.H.R., supervised research and coordinated the study. I.T, A.G, S.F, S.J.J, R.H, R.D, A.G, L.B, K.D, O.O, R.A, H.D, A.B, N.H.R., M.D provided clinical data and DNA specimens. P.J.C. and E.P designed the sequencing strategy. A.G., and E.P. coordinated sample acquisition. Y.T, J.E.A.O., Y.Z, Y.P, M.F.L, D.L. A.B and E.P, performed bioinformatic analysis. A.G, L.B, K.D, H.D, clarified validation cohort queries. Y.T, J.A.O, Y.Z, with input from M.F.L and E.B developed web-based clinical decision support tool. Y.T. and E.P. prepared figures and tables. Y.T, S.M.D, B.J.P.H and E.P. wrote the manuscript. All authors reviewed and approved the manuscript during its preparation.

## Competing interests

E.P. is a founder, equity holder and has a fiduciary role in Isabl, a cancer whole genome sequencing analytics company. The remaining authors declare no competing interests.
