## [Peer Review File · Nature Communications]

Title: Unified classification and risk-stratification in Acute Myeloid LeukemiaEditorial Note: This manuscript has been previously reviewed at another journal that is not operating a transparent peer review scheme. This document only contains reviewer comments and rebuttal letters for versions considered at *Nature Communications*.

REVIEWERS' COMMENTS

Reviewer #1 (Remarks to the Author):

The authors have satisfactorily addressed the prior critiques.

Reviewer #3 (Remarks to the Author):

The authors have largely addressed the concerns of Reviewer 2. However, there is one point points to clarify, regarding the Reviewer comment: "A concern bothers me that when talking about the mutations. The authors wrote that many of the patients were intensively treated. When were the biopsy were drawn? Would the molecular profiles of these patients be those before or after they were treated? There might be a lineage selection process during the patient journey."

The authors confirm that molecular assessment of UK-NCRI is taken at diagnosis, but what about the validation AMLSG cohort? Also, the reference for AMLSG is not given properly (ref 8 is not correct), please provide.

Reviewer #1 (Remarks to the Author):

The authors have satisfactorily addressed the prior critiques.

Response: Thank you.

Reviewer #3 (Remarks to the Author):

The authors have largely addressed the concerns of Reviewer 2. However, there is one point points to clarify, regarding the Reviewer comment: "A concern bothers me that when talking about the mutations. The authors wrote that many of the patients were intensively treated. When were the biopsy were drawn? Would the molecular profiles of these patients be those before or after they were treated? There might be a lineage selection process during the patient journey."

The authors confirm that molecular assessment of UK-NCRI is taken at diagnosis, but what about the validation AMLSG cohort? Also, the reference for AMLSG is not given properly (ref 8 is not correct), please provide.

Response: We thank the reviewer for the comment.

The authors would like to confirm that all samples were diagnostic, treatment naive for both the UK NCRI and AMLSG Cohorts.

We have now added that the molecular assessment for the validation AML SG cohort was also at diagnosis:

"Data from 1,540 AML patients from the AML-SG¹⁰ (median age = 50) with comparable molecular annotation at diagnosis were used as a validation cohort (S.Table 1, S.Data 1, S.Figure 1)."

We thank the reviewer for their comment about the AMLSG reference. It has now been updated correctly:

Data from 1,540 AML patients from the AML-SG¹⁰

¹⁰Papaemmanuil, E., Döhner, H. & Campbell, P. J. Genomic Classification in Acute Myeloid

Leukemia. *The New England journal of medicine* vol. 375 900–901 (2016).